# Psychosocial Processes in Healthcare Workers: How Individuals’ Perceptions of Interpersonal Communication Is Related to Patient Safety Threats and Higher-Quality Care

**DOI:** 10.3390/ijerph20095698

**Published:** 2023-05-01

**Authors:** Johanna Elisa Dietl, Christina Derksen, Franziska Maria Keller, Martina Schmiedhofer, Sonia Lippke

**Affiliations:** 1Health Psychology and Behavioral Medicine, School of Business, Social and Decision Science, Constructor University, 28759 Bremen, Germany; jdietl@constructor.university (J.E.D.);; 2Klinikum Bremerhaven Reinkenheide gGmbH, Treatment Center for Psychiatry, Psychotherapy and Psychosomatic, 27574 Bremerhaven, Germany

**Keywords:** interpersonal communication, social relations, psychological safety, patient safety

## Abstract

Interpersonal communication, as a central form of social resource derived from social relations, is crucial for individuals coping with threats in the workplace, especially for hospitals that provide high-quality care and patient safety. Using social system mentalization as a theoretical background, we applied psychosocial processes and a psychodynamic system approach to get insights on how healthcare workers interact with team members and patients. The goal was to test the following hypotheses: H1: Better communication is associated with fewer patient safety threats (H1a) and higher-quality care (H1b). H2: The associations between communication and patient safety threats (H2a) and higher-quality care (H2b) are mediated by psychological safety. In this two-studies design, we conducted a cross-sectional hospital survey (*N* = 129) and a survey of obstetric team members (*N* = 138) in Germany. Simple mediation analyses were run. Results revealed that communication is associated with safety performance. Further, the mediating effect of psychological safety between communication and safety performance was demonstrated. These findings contribute to an understanding of social relation representations, as individuals’ communication interrelates with safety performance mediated by psychological safety to complement healthcare and public health strategies. With a better understanding of communication and psychological safety, tools, routines, and concrete trainings can be designed.

## 1. Introduction

Intrapersonal resources and interpersonal communication are central for social bonds, group membership or social relations, and coping with different workplace threats [1,2]. Scholars have shown the importance of organizational environments impacting individuals’ work behavior and attitudes. More precisely, team members’ perceptions are rooted in social interactions; therefore, it is necessary to take an individual’s representations of organizational contexts (e.g., perceptions of colleagues, customers, or work relationships) into account to understand organizational functioning [3].

Hospitals are high-reliability organizations (HROs) that must deal with increased complexity in highly demanding, stressful, and dangerous circumstances [4]. Accordingly, individual coping with different threats in the workplace and organizational functioning in healthcare is key to providing safe patient care. Especially in obstetrics, high requirements, e.g., teamwork challenges, lead to difficult working conditions and psychological stress [5,6]. Interprofessional and interdisciplinary teams, such as in obstetrics, face complex interaction patterns due to their need to integrate different educational backgrounds and work approaches [5,6]. To gain a better understanding of the terminology used when discussing patient safety and potential threats or risks, it is necessary to differentiate between what is a threat and what constitutes a risk. According to a common understanding in healthcare settings, threats to patient safety refer to possible preventable adverse events that could endanger patients’ well-being and health status. Threats paired with organizational vulnerabilities, such as ineffective controls, human errors, or low psychological safety, may lead to potential risks for the institution, organization, or healthcare system as a whole due to legal penalties, financial losses, or damage to reputation. Hence, we postulate that threats in the form of preventable adverse events put overall patient safety at risk, e.g., by treatment delay due to ineffective and insufficient communication leading to potential malpractice [7].

Human failures and, particularly, communication errors are considered the main source of patient harm [8,9]. Thus, hospitals increasingly depend on collaborative work and human interaction to achieve safety goals, e.g., (perceived) patient safety and high-quality care [10,11,12]. In line with the psychosocial processes and social learning theory [13], human interactions are essential for quality relationships, improved knowledge, and a shared understanding of events (“shared reality”) and shared mental representations [14,15,16]. Therefore, enhancing social relations at work (reflecting mental models of quality work relationships) is crucial to function effectively and avoid failures, corresponding to patient safety (as with global health threats) [11,17].

In line with Petriglieri and Petriglieri [18], a systems psychodynamic approach fosters new organizational insights into how individuals and their own needs, thoughts, fears, or desires interact with team members to constitute collectives and meaning in their work environment. This study, therefore, investigates interpersonal communication and psychosocial processes, specifically in the application-oriented context of healthcare and patient safety.

Communication is directly linked to cognitive products such as attributions or perceptions that need to be verbally shared with others [19]. A mutual understanding can only be achieved through interpersonal communication and the creation of meaning. Nevertheless, mutual understanding is not always achievable, which may lead to misunderstandings [20]. This was consistently found in prior research on communication deficits leading to patient safety threats [8,9]. The construction of meaning is a social process in which interaction partners (e.g., team members or patients) must adjust to each other’s perspectives to create a shared reality [20,21]. This aligns with the concept of interpersonal communication in healthcare for both patient–provider and team communication [22]. Hence, communication in a shared reality (department or workgroup) generates a collective mental model [20]. Consequently, effective interpersonal communication in healthcare might not directly translate to effective collaboration but work via shared beliefs such as psychological safety.

There are practical examples of interpersonal communication as a central form of social resource in the current patient safety literature that identify techniques to create clear and sufficient communication. For instance, practical examples of interpersonal communication as a central form of social resource in hospitals are the following:Doctor–patient communication: One of the most important forms of interpersonal communication in hospitals is the interaction between medical doctors/physicians and patients. Effective communication between doctors and patients can help build trust, alleviate anxiety, and improve patient outcomes. For example, a physician might explain a medical diagnosis or treatment plan to a patient clearly and empathetically, which can help the patient feel more comfortable and confident in their care, help compliance, and help them be satisfied with the treatment outcomes [23,24,25].Nurse–patient communication: Nurses also play a critical role in patient care, and effective communication between nurses and patients is essential for providing high-quality care. Nurses might use interpersonal communication skills to listen to patients’ concerns, explain medications or treatments, or provide emotional support during difficult times [23,24,25].Interdisciplinary team communication: In hospitals, interdisciplinary teams of healthcare professionals often work together to provide coordinated care to patients. Effective communication between team members can help ensure patients receive appropriate care and avoid unnecessary complications and confusion [23,24,26].Communication with accompanying persons such as partners and family members: Hospitalization can be a stressful and emotional experience for patients and their families. Effective communication between healthcare providers and accompanying persons can help ease anxiety and improve patient outcomes. For example, a HCW might communicate with a patient’s family about their loved one’s condition, treatment plan, or discharge instructions [24,25,27].

When thinking about obstetric care, there are even more specific examples:Prenatal care provider–patient communication: Effective communication between the obstetric care provider and the patient is essential to ensure the health and well-being of the mother and baby. During prenatal visits, the obstetrician or midwife might use interpersonal communication skills to discuss the pregnancy’s progress, address any concerns, and explain what to expect during the next stages of pregnancy [24,27].Labor and delivery communication: Communication between the obstetric care provider and the patient is crucial during labor and delivery. Effective communication can help the patient feel supported and informed throughout the delivery process. The obstetrician or midwife might use interpersonal communication skills to explain the different stages of labor, provide pain management options, and help the patient make informed decisions about the delivery [24,25,27].Postpartum care provider–patient communication: Communication between the obstetric care provider and the patient is still important after delivery. The care provider might use interpersonal communication skills to provide instructions on postpartum care, discuss breastfeeding or formula feeding options, and address any concerns or complications that arise [24,25,27].Obstetrician–midwife communication: Often, an obstetrician and a midwife work together to provide care to a patient during pregnancy, labor, and delivery. Effective communication between the two professionals ensures that the patient receives consistent and coordinated care. The obstetrician and midwife might use interpersonal communication skills to discuss the patient’s medical history, provide updates on the pregnancy´s progress, and collaborate on decisions regarding the delivery [24,26,27].Nurse–physician communication: Nurses and physicians often work together in obstetric care to provide comprehensive care to patients. Effective communication between nurses and physicians is important to ensure that the patient’s needs are met and that the care provided is safe and effective. For example, a nurse might communicate with a physician about a patient’s vital signs or symptoms, or a physician might collaborate with a nurse on the patient’s care plan [24,26,27].

Of course, there are many more combinations and instances, constellations, and risks for misunderstandings but also options for improving and preventing error. Communication tools and strategies, such as debriefings [28], speaking up [29,30], or structured handovers (ISBAR) [31] are critical hospital communication structures. Debriefings are reflections and discussions after patient care or a medical procedure to identify improvements or challenging factors to provide safer care and team performance [28,32]. Speaking up is the ability to challenge authority by expressing concerns and opinions in difficult situations. It is a substantial and highly difficult aspect of communication to foster patient safety. Structured handovers (e.g., using the ISBAR technique) are the standardization of patient information transitioning and hence an essential tool to ensure correct information exchange [33] and thus create a shared understanding.

Nevertheless, there is more to effective and safe communication in healthcare than applying technical communication skills. As stated above, a mutual understanding of critical situations is crucial for collaboration and effective teamwork [31]. Individuals exposed to (emotionally) stressful environments must feel safe and appreciated by their team members to cope with challenging social relationships at work [34], especially for HCWs. HCWs are at risk of perceiving other team members as threatening due to the hierarchical and multidisciplinary organizational structures in their work environment [11,12,35]. To ensure the quality of care, team members need to feel they can report mistakes, propose new ideas, or seek feedback without fear of damaging their self-esteem, status, or career due to relationship threats. This refers to the aspect of “psychological safety” [34,36]. In line with Newman et al. [37], psychological safety is a mental representation that fosters learning processes and work performances (e.g., safety performance indicators). Edmondson [38] described psychological safety as an interpersonal interaction belief. Team members relate to significant others (e.g., colleagues) through social signs, which affect their perceptions and expectations. Therefore, considering individual perceptions of the context of self and others could broaden our understanding of social relations and social resources [3,39]. In line with shared reality, team members tend to have similar perceptions about psychological safety because they are exposed to the same work environment and shared experiences [6,38], which shapes their perceptions and mental representations [40]. Thus, psychological safety as a shared belief could foster an individual mental representation of being safe to take risks in the team context.

Individuals’ perceived psychological safety can be seen as a critical mediator that may transfer the safety performance implications associated with communication. Psychological safety significantly affects learning behaviors, such as engagement and performance [10,41,42]. The same applies to HCW and patient–provider communication [8,9]. Hence, a positive association between interpersonal communication, psychological safety, and patient safety outcomes can be assumed. Nevertheless, the link between interpersonal communication, psychological safety, and patient safety outcomes is not sufficiently understood, neither theoretically nor empirically [42].

Extensive patient safety literature has investigated factors in social interaction and team interventions to improve collaboration and communication to foster patient safety and care. Important training programs, such as TeamSTEPPS, have been developed to improve communication and teamwork and reduce errors [43]. Thus, many studies in the healthcare context examine team relationships and their effects on communication, coordination, or leadership [44]. Nevertheless, few studies have investigated communication as a central social resource, as most studies that have evaluated or trained communication included in a higher teamwork construct or regarded it as a technical skill rather than an interpersonal construct.

As a result, the association between communication errors and patient safety has been much studied, yet the *psychological mechanisms* of this relationship have not been clearly explored. However, research into this mechanism is important to derive further patient safety measures and to improve social resources such as communication and collaboration. Integrating an applied framework of psychosocial processes and mentalization can help to understand the link between communication, psychosocial processes, and patient safety in the healthcare setting of obstetrics. From a systems psychodynamic perspective, mentalization refers to the ability to interpret and predict behavior by attributing mental representations about feelings, attitudes, or desires [45].

Di Stefano et al. [46] propose a general mentalization model for organizations and team contexts, which focuses on mental representations with respect to intersubjective experiences of belonging environments, particularly to work organizations. Their framework is based on the concepts of mentalization and reflective functioning and assumes a process of constructing shared meaning by considering self and others. Thus, their mentalization theory of social systems considers self and self–other dynamics in the work environment. More precisely, the mental states underlying the behavior of team members are essential to understanding and generating adequate social interactions and interpersonal relationships in the work context [46].

Social system mentalization is the perception and interpretation of behavior not only facing other individuals but underlying relational dynamics. Therefore, individuals require an examination of actions and their own experiences to draw conclusions about the cognitive and affective states of others to find suitable actions and steering possibilities in the workplace. Based on the applied mentalization concept, team members should be capable of mentalizing others’ feelings, beliefs, and intentions while also attributing meaning to their feelings and emotions resulting from the work environment [46].

A typical instance in healthcare that illustrates the need for adequate mentalization to understand effective interaction concerns psychological safety [10,36,47]. From a social constructionist perspective, to understand social relationships at work and consistent with the mentalization framework, psychological safety is not generated out of a vacuum. It rather develops from social interaction and communication [46,48,49]. HCW and patient–provider communication is the basis for effective patient care and, consequently, patient safety [50] and is usually framed as an interpersonal process, which is based on a shared understanding between care providers [22] resulting in the possibility of sharing a perception of psychological safety and thus delivering high-quality care [51,52]. Additionally, psychological safety requires mentalization and cognitive anticipation and reviewing others’ responses to interactions in communication [47,53]. HCW and patient–provider communication can therefore be regarded as an antecedent of psychological safety, which is a key explanatory mechanism linking interpersonal communication with safety performance indicators.

This hypothesized link is consistent with an individual interpretation of the input–process–output model of team effectiveness (IPO), which is a systems theory explaining how specific factors interact with each other to result in team performance [54,55]. The IPO model can be integrated with the mentalization theory. Therefore, communication can be understood as an input since it can be perceived as a team characteristic. An individual’s cognitive process, on the IPO model, is the formation of a shared belief, in this case psychological safety. Finally, both the input and the underlying mechanism via social processes lead to the (safety performance indicator) output of high-quality care and perceived patient safety [54,55]. Research has depicted that resuscitation teams share a mental model (process) based on their training and communication (input), which implies a cognitive understanding of each other’s roles and goals in the emergency care process with the final aim of effective emergency response (output) [56]. Thus, perceived psychological safety can be assumed to be a key mediator between interpersonal communication and safety performance indicators in the context of the IPO model.

Based on mentalization [46] and the IPO model [54,55] this study aims to investigate psychosocial processes and a systems psychodynamic perspective on perceived interpersonal communication in healthcare, first in a general hospital setting focusing on HCW perception of communication with patients and second in an obstetric, interprofessional team member context, centering perceived communication within the team and the patients.

The main *research question* explores how interpersonal communication, psychological safety, and safety performance indicators (i.e., patient safety threats or quality of care) are interrelated. The research is based on assessing individual perceptions, because individuals cope with different threats in the workplace and subjective perceptions are important to understand shared reality and mental representations. Shared reality is connected to individual experience in terms of perceiving it as real or truthful [15], similar to the individual level of psychological safety research. This also stresses the degree to which team members feel interpersonally safe or non-threatened [36,42]. Therefore, the individual level of analysis provides important new insights into social interactions, shared psychological safety, well-being, and health.

Hence, we hypothesize:

**H1:** 
*Better interpersonal communication is associated with fewer patient safety threats (H1a) and higher-quality care (H1b).*


**H2:** 
*The associations between interpersonal communication and patient safety threats (H2a) as well as higher-quality care (H2b) are both mediated by psychological safety.*


The present research adopted a two-study design to replicate results with two samples. Study 1 was conducted as a cross-sectional hospital survey. Study 2 aimed to replicate the results of Study 1 in the organizational, interdisciplinary field setting of obstetrics. Therefore, we expect the hypotheses proposed as part of Study 1 to be supported and confirmed by results from Study 2, relying on a sample of obstetric HCWs from two representative university hospitals in Germany.

## 2. *Study 1* Materials and Methods

In the first study, an online survey was conducted. We assessed how HCWs perceive interpersonal communication with patients, psychological safety, and safety performance indicators in their belonging environment of patients and team members. The study was performed within a research project concerning digitally supported communication in obstetrics and gynecology. The project was funded by the German Innovation Fund of The Federal Joint Committee (G-BA) [57].

### 2.1. Participants and Procedure

In an online survey, employees at German hospitals were recruited (Table 1). The online survey was distributed via press releases, professional associations, healthcare social media groups, and an E-mail list with quality management representatives from 1500 hospitals with a request to forward the survey to their employees.

Potential participants were informed about the study aim and data security, then indicated their informed consent before participating in the survey. Individuals were included in the analysis if they were over 18 years and worked or were undergoing vocational training at least part-time in healthcare or a related setting, e.g., physicians, nurses, or midwives.

Approval was given by the ethics committee at the participating university (dated 17th September 2019). Data were collected from 9th October 2019 to 6th March 2020. After the questionnaire was pilot tested with three test participants, 173 individuals initially completed the questionnaires, of which 43 were excluded because of a high proportion of missing data in crucial scales. We further excluded one participant who rated every single item with the same value (highest possible). Therefore, the final sample consisted of *N* = 129 HCWs, who took part in the study on a voluntary basis. A detailed overview of socio-demographic data is provided in Table 1.

### 2.2. Measures

Participants provided self-reported data concerning perceived interpersonal communication with patients, perceived psychological safety, socio-demographic data, and safety performance indicators, which was operationalized as perceived patient safety threat and quality of care. Participants rated most items using a six-point Likert scale ranging from ‘1’ (absolutely not) to ‘6’ (absolutely). All items for each construct were aggregated in terms of a mean score.

*Psychological safety.* We measured perceived psychological safety using Edmondson’s [36] adapted four-item measure. Sample items include, “When someone on my team makes a mistake, it is often used against them” or “Working with members of this team, my unique skills and talents are valued and utilized” (Cronbach’s alpha = 0.77).

*Interpersonal communication with patients.* We measured communication based on Rider and Keefer’s [58] interpersonal communication competencies. The eight-item scale was developed by reviewing an initial item pool with healthcare experts [58]. A sample item is “I explain all examinations or procedures in such way that my patients understand them.” Cronbach’s alpha was 0.82.

*Patient safety threats.* We assessed safety performance indicators regarding threats, such as the perceived preventable adverse trigger scale. We adapted a patient-centric trigger-for-adverse-events scale from Keller et al. [59] to the group of HCWs. We measured how often team members noticed possible patient safety threats using a 15-item scale with the sample item “Colleagues or I had insufficient knowledge of technical equipment” Cronbach’s alpha was 0.93 [59].

*Quality care.* Based on patient safety and quality-of-care assessment literature, we constructed a twelve-item scale, e.g., “I believe that errors and complaints are handled responsibly in our hospital”, Cronbach’s alpha was 0.83 [60,61,62].

We applied a strict socio-demographic security requirement to ensure response rates through greater anonymity. Therefore, we assessed socio-demographic variables regarding sex, age, and profession as categorical data, with the “I’d rather not say” option for participants who felt reluctant to provide socio-demographic information. Age and profession were measured in four categories (for age: “younger than or 25 years”, “26–40 years”, “41–55 years”, “56 years or above”; for profession: “physician”, “midwife”, “nurse”, “other”). Sex was divided into three groups (“men”, “women”, “diverse”).

### 2.3. Data Analysis

All data analysis was conducted using IBM SPSS Version 27. Bivariate Pearson correlation coefficients were calculated for all variables to test for intercorrelations. Two mediation analyses were conducted to examine the association between self-reported interpersonal communication (communication) and perceived patient safety threats as well as communication and quality of care, with psychological safety as the mediator.

The Baron and Kenny approach was applied along with a direct test for the indirect effect via bootstrap analyses using 5000 resamples by applying the Process Macro Model 4 for SPSS version 3.4 [63]. HCWs differ in work approaches, duties (e.g., midwives conduct births with a salutogenic model and physicians are led rather by pathological birth processes), and educational background. Further, the degree of responsibility also differs according to position or age [5,17,26].

Therefore, as we assumed that professional experience, hierarchy, and gender may be associated with the HCWs’ communication and interaction patterns, we controlled for age, sex, and profession. Sex, age, and profession were consequently added as dummy-coded covariates and were adjusted for all independent and mediator variables [63]. For age, “younger than or 25 years” was chosen as the reference group. Concerning profession, “physicians” was used as the reference group. Sex was included as a binary variable due to no diverse participants.

## 3. *Study 1* Results

Descriptive statistics and intercorrelations among variables are reported in Table 2.

### 3.1. Testing H1: Better Interpersonal Communication Is Associated with Fewer Patient Safety Threats (H1a) and Higher-Quality Care (H1b)

Communication was positively related to psychological safety and negatively associated with patient safety threats (matching H1a, Table 2). Communication was positively related to quality of care (supporting H1b, Table 2).

### 3.2. Testing H2: The Associations between Interpersonal Communication and Patient Safety Threats (H2a) as Well as Higher-Quality Care (H2b) Are Both Mediated by Psychological Safety

To examine H2, mediation analyses were conducted to test whether the association between self-reported interpersonal communication (communication), perceived patient safety threats (threats, Figure 1), and higher-quality care (care, Figure 2) is mediated by perceived psychological safety (psychological safety) when controlling for the covariates sex, age, and profession.

Covariates were not significant at *p* < 0.05. Communication was not associated with threats directly (γʹ = −0.142, *p* = 0.093), but revealed a total standardized effect (γ_1_ = −0.252, *p* = 0.009). Communication was associated with psychological safety (α_1_ = 0.218, *p* = 0.025). Furthermore, psychological safety was significantly associated with threats (β_1_ = −0.502, *p* < 0.001). Lastly, bootstrapping procedures using 5000 resamples revealed a significant standardized indirect effect of communication on threats mediated by psychological safety (α_1_*β_1_ = −0.109, 95% CI −0.22, −0.01]), supporting Hypothesis 2a. Overall, 34.9% of the threat variance could be explained by communication and psychological safety.

To test Hypothesis 2b, a mediation analysis with care as the outcome variable (see Figure 2) was performed. Communication was positively linked to care with a total standardized effect (γ_2_ = 0.286, *p* = 0.002) and a smaller but still-significant direct standardized effect (γ_2_ʹ = 0.194, *p* = 0.024). Communication was significantly associated with psychological safety (α_2_ = 0.218, *p* = 0.025). Furthermore, psychological safety was positively associated with care (β_2_ = 0.423, *p* < 0.001).

Hypothesis 2b, which stated that the association of communication with care is mediated by psychological safety, was thereby supported (α_2_*β_2_ = 0.092, 95% CI [0.01, 0.20]). Overall, 34.0% of the variance in care could be explained.

## 4. *Study 2* Materials and Methods

The second study was also part of a research project concerning digitally supported communication in obstetrics and gynecology and was funded by the German Innovation Fund of The Federal Joint Committee (G-BA) [57].

### 4.1. Participants and Procedure

A sample of *N* = 138 interprofessional team members was recruited from two German obstetric university hospitals with approximately 2800 to 3200 deliveries annually and affiliated neonatal intensive care units. Ethical approval for cross-sectional data collection at the obstetric hospitals was granted as part of the research project’s ethical approval from the two hospital ethics committees. Participants were interprofessional and interdisciplinary team members from the two hospitals, who voluntarily participated in the study. The research project aimed to conduct a full survey; therefore, the majority of the sample (*N* =138 out of *N* =141) who worked at the hospitals were participating in the research project. They were informed by the on-site researchers about the research project in person and were given written information with contact details and consent forms. In case they required further information and clarification, they were invited to contact the on-site researchers.

All included participants gave written informed consent for participation. Participants were eligible if they were over 18 years and had worked at least part-time in any obstetric unit or in a gynecological unit affiliated with the delivery rooms. HCWs under training were also included. Data were collected at both hospitals simultaneously from 2nd January to 15th March 2020. A detailed overview of socio-demographics is provided in Table 3.

### 4.2. Measures

Psychological safety. Psychological safety was measured with the same version of Edmondson’s [36] adapted four-item scale used in Study 1. Cronbach’s alpha was 0.71.

Interpersonal communication within the team and with patients. We measured interpersonal communication with an adapted seven-item scale from our measure in Study 1. To assess both the understanding of one’s own and other’s perception, we rephrased the items from a self-perspective of communication with patients to another perspective reflecting the shared reality of communication, with the sample item “We as a team take the amount of prior knowledge of the patient and how much she can understand into account.” [58]. Cronbach’s alpha was 0.84.

Safety performance indicators for patient safety threats and quality care. We assessed safety performance indicators with a trigger scale for threats and a care scale, as in Study 1. Cronbach’s alpha for the threats scale was 0.91 [59]. We adapted a short seven-item scale measure of care from Study 1 for practicality. Additionally, we omitted one item due to poor parameters; therefore, Cronbach’s alpha for care was 0.67 [61,62].

Demographic variables were assessed in the same fashion as in Study 1 (see Table 3). Due to comparability issues and security requirements (e.g., only a few male midwives), we assessed socio-demographic variables regarding sex, age, and profession as categorical data with the option “I’d rather not say” to secure that no conclusions can be drawn about the participants. Age and profession were measured in four categories (for age: “younger than or 25 years”, “26–40 years”, “41–55 years”, “56 years or above”; for profession: “physician”, “midwife”, “nurse”, “other”). Sex was categorized into three groups (“men”, “women”, “diverse”).

### 4.3. Data Analysis

All data were analyzed as in Study 1, to replicate previous mediation analyses.

## 5. *Study 2* Results

Descriptive statistics and intercorrelations among variables are reported in Table 4.

### 5.1. Testing H1: Better Interpersonal Communication Is Associated with Fewer Patient Safety Threats (H1a) and Higher-Quality Care (H1b)

In line with Hypothesis 1a, perceived interpersonal communication (communication) was positively related to psychological safety and negatively related to threats. Corroborating Hypothesis 1b, communication was positively related to care.

### 5.2. Testing H2: The Associations between Interpersonal Communication and Patient Safety Threats (H2a) as Well as Higher-Quality Care (H2b) Are Both Mediated by Psychological Safety

To investigate Hypothesis 2, two mediation analyses were conducted to examine the link between communication and threats and to examine whether care was mediated by psychological safety while controlling for age, sex, and profession. The covariates showed no significant associations. Unstandardized coefficients for the mediation analysis on perceived patient safety threats in the obstetric HCW sample are reported in Figure 3.

The analysis revealed a total standardized effect (γ_3_ = −0.309, *p* < 0.001) and a smaller direct standardized effect (γ_3_ʹ = −0.213, *p* = 0.015) between communication and threats. Communication was associated with psychological safety (α_3_ = 0.268, *p* = 0.003), and psychological safety was significantly associated with threats (β_3_ = −0.359, *p* < 0.001). Corroborating Hypothesis 2a, the mediation was significant, as Bootstrap analyses using 5000 resamples showed a significant standardized indirect effect of communication on threats (α_3_*β_3_ = −0.096, 95% CI [−0.18, −0.03]). Overall, 26.7% of threat variance could be explained by communication and psychological safety.

For care, unstandardized coefficients in the obstetric HCW sample are reported in Figure 4, which presents the results for Hypothesis 2b. The mediation analysis revealed a total standardized effect (γ_4_ = 0.376, *p* < 0.001) and a smaller, still-significant direct standardized effect (γ_4_ʹ = 0.310, *p* < 0.001) between communication and care. Communication was significantly associated with psychological safety (α_4_ = 0.268, *p* = 0.003). Additionally, psychological safety was significantly associated with care (β_4_ = 0.249, *p* = 0.004). The bootstrap analysis revealed a significant indirect effect (α_4_*β_4_ = 0.067, 95% CI [0.01, 0.15]). Overall, 28.8% of the variance of care could be explained.

## 6. Discussion

The current study aimed to investigate interpersonal communication as a central form of social resource derived from social bonds, group memberships, or social relations. Therefore, HCWs’ perception of interpersonal communication and the mediating role of psychological safety between communication and safety performance indicators was researched. The results suggest that interpersonal communication is crucial for social resources and relations in HROs (high-reliability organizations such as hospitals) to prevent global health threats (patient safety threats) and maintain health and well-being (quality care).

The study responds to repeated calls for new insights into the understanding of the health and well-being of employees, as well as patient care and patient safety threats, by understanding the psychological mechanisms of social resources and interpersonal communication to complement the strategies of healthcare and public health [1,18,64,65]. By integrating social resources resulting from social relations as well as psychosocial processes such as social system mentalization [46] with the theoretical foundation of the IPO model [54,55], the present research demonstrated that psychosocial processes (psychological safety) represent an underlying mechanism in the relationship between perceived HCW and patient–provider communication (input) and perceived safety performance indicators (output) [54,55].

We specifically found that psychological safety mediates the association of interpersonal communication with quality of care and patient safety. We tested our hypotheses using an individual-level two-study design. Study 1 used an online HCW sample from different healthcare areas. In the second study, with obstetric HCWs, we replicated the results from the first study with a focus on the shared reality of communication (of patients and team members). Our findings are in line with previous research findings that the interpersonal skill to mentalize another’s perspective is linked to improved team outcomes, e.g., patient safety [48]. Individuals learn and adapt by observing and interacting with others, whereby social and accepted norms manifest and individual social representations are formed, which are crucial for social relations and social resources [66,67].

### 6.1. Integration of Results

Our research aims to progress the understanding of social phenomena in healthcare, concentrating on the interplay of psychosocial processes and cognitive and emotional factors by embedding our data into a systems psychodynamic framework. Our study teases out the importance of psychological safety as such a facilitator or barrier, and it also adds a perspective on what drives mentalization and the importance of the social context. Therefore, we analyzed individuals’ perceptions of the social context, which stresses the construction process of a (shared) work reality [40].

Previous work has demonstrated that a supportive work context is positively related to psychological safety [42]. Considering individual reports of HCW and patient-provider communication in the well-studied relationship between psychological safety and performance outcomes (quality care and patient safety threats; [42]), one can target such social relationships at work [48]. Healthcare, as a high-reliability organization (HRO), requires introspection and social skills, e.g., communication, to efficiently deal with sophisticated relations with patients and colleagues. Mentalization is the ability to think and reflect on one’s own and other´s feelings, desires, or needs, which underlie one’s own and other´s (communication) behavior [68,69]. Communication between HCWs and patients could be characterized as an interaction between informational and implicit emotional exchange processes creating relationships. Perceived communication leads to attributional assessments of others on an individual, team, or organizational level [70], manifested in perceived psychological safety. Our results offer further support to previous findings [71], suggesting that quality interactions in terms of interpersonal communication with patients and colleagues are important to nurture and cultivate perceptions of psychological safety based on mentalization [68,69].

Psychological safety is a belief based on shared experiences, which could develop through specific interactions and conversational behavior. Further supported by the IPO model, our mediation model indicated that communication (as an input/team characteristic) is related to patient safety outcomes through psychological safety as an intervening mechanism [10,48,54,55]. Given that psychological safety as an interpersonal concept builds on interactions [10], we show that communication fosters the formation of a shared reality. In emphasizing psychosocial processes and interpersonal communication, our results are also consistent with social system mentalization, which defines the process of generating shared meaning in the work context.

If social resources are derived from social bonds, group memberships, and social relations, as well as mentalization, and if these are lacking in a social system such as healthcare, individuals are exposed to significant psychosocial threats [46]. Poor interpersonal communication may outline the deficiency-reflective competence of interactive work experience, negatively impacting individual job perceptions and team member relationships in terms of psychological safety, which interferes with sustainable work [46]. Therefore, psychological safety creates an atmosphere that aids team members in avoiding mistakes [47] as well as promotes health and well-being. This is supported by our findings that psychological safety is associated with lower perceived patient safety threat and higher-quality care. This investigation opens new conceptual ground for research on systems psychodynamics and how shared mental models affect social (health) behavior in HRO contexts.

Work overload and excessive demands are common in the inpatient healthcare system [72,73]. Both of our studies were conducted with individuals working in healthcare settings, which are prone to the development of challenging situations. Our findings underscore the importance of interpersonal communication and mentalization to deliver safe care. Strengthening individual communication skills, including empathy, through training to target mentalization could enhance psychological safety, which is key in ensuring overall high quality of care as well as patient safety, as shown by the mediation analyses of the present study. As depicted in the mediation analyses, greater psychological safety was associated with greater care and reductions in threats to patient safety. Therefore, our results suggest that it is imperative for healthcare organization leaders and medical executives to acknowledge that a limited consideration of HCWs psychological safety could have the negative effect of adversely impacting patient safety. Hence, this work encourages organizations to foster psychological safety by training communication competencies and creating an environment for healthcare proivders to be open to questions, feedback, and concerns.

Our results indicate that psychological safety should be an essential target within interventions such as team trainings in healthcare. Moreover, technical communication tools (e.g., closed-loop communication, debriefings, speaking up, and structured handovers) could be integrated into the concept of interpersonal communication by addressing interpersonal communication and its challenges and tools. Therefore, interpersonal communication could be framed and understood as an important social resource to foster a favorable environment for communication and for the exercise of quality in intersubjective relationships.

Our research extends previous work, which examined effective communication in promoting patient safety or team performance in healthcare settings, by suggesting psychological safety is essential in relation to effective communication, leading to better patient outcomes. By investigating communication as a central social resource (not as a subdimension of teamwork, or conceptualized primarily as a technical skill, rather than a social resource), we explored psychosocial mechanisms regarding how HCWs’ perceptions of communication are related to patient safety threats and quality care. Research into the mechanisms of perceptions of social resources in terms of communication and patient safety is key as it reveals partial and full mediation models. Therefore, we provide significant insights into psychosocial processes, which could help to further develop patient safety measures and to improve social resources such as communication and collaboration.

One promising approach is the systematic training of healthcare provider communication [74] as an input to start an enhanced process. However, sustainable improvements will only be achieved if the training is embedded along with an enhanced safety culture within the organization [75]. Top-down strategies to implement patient safety in healthcare settings are a key target. In addition, participatory or co-creative development and the delivery of necessary resources have been reported to be beneficial [76,77].

### 6.2. Limitations of the Current Research and Suggestions for Future Studies

When interpreting the results, several limitations need to be considered. One of the main limitations of this study is the lack of validated scales. Scales were either adapted or newly constructed due to the lack of scales in previous literature (see Appendix A). Hence, there may be reduced reliability for certain proposed constructs, especially for quality care in Study 2. Therefore, future studies should evaluate the psychometric properties of the newly developed and adapted scales in the context of healthcare.

This study investigated individuals who were part of changing teams; therefore, we examined individual perceptions rather than teams. All variables evaluating the proposed constructs were (retrospective) self-report measures collected cross-sectionally. Although the individual perspective is clearly important to understand the construction of shared reality and social context, the associated limitations of self-reported measures, such as common method biases (recall bias or social desirability), need to be acknowledged when interpreting responses. To overcome this limitation, analysis of psychological safety, interpersonal communication, and safety performance indicators as group-level constructs or by means of observation is needed to get a better understanding of others and team dynamics. Moreover, an investigation with a qualitative study in multidisciplinary healthcare groups could provide new insights into the relationship between interpersonal relationships and communication, as we have done successfully before (e.g., [27]), but which should also be done more often in the future.

The two studies were correlational only. Thus, future research should employ time-lagged, longitudinal, or experimental designs using objective measures. Currently, only interrelations may be postulated without establishing causal effects. Measuring patient safety and quality of care by incident reporting systems and analysis of routine data or assessments of patients of the accompanying team could provide further insights. Regarding the IPO model, research designs and analyses are needed to determine the underlying processes as dynamic changes over time.

Another important research question will regard the development of psychological safety in individuals and how this stems from perceived interpersonal communication as well as how they align with other team members’ perceptions. A mixed-methods approach integrating observational measures and interviews could give further insight.

Other limitations regard the lack of representability of the sample in both studies; hence, limited generalizability to other healthcare sectors needs to be acknowledged. Despite the comprehensive recruitment strategy in Study 1, only a small number of HCWs could be included in the final analyses, suggesting a certain selection and thus sampling bias. In Study 2, the university hospitals providing the highest level of care were pre-selected as they took part in a research project. Together with the rather limited sample size, the results from this study might only be generalizable to social relationships at work within obstetrics. However, the study design had two representative hospitals from Germany participating in the study. Thus, the findings should generally be generalizable to other hospitals. Nevertheless, within these hospitals, there might have been a selective sample, and future research is needed to test this. To replicate findings in other work-related areas in different healthcare contexts, more research is needed.

### 6.3. Implications for Practice

Our findings indicate that training HCWs in interpersonal communication is important to build social resources and create a shared belief that interpersonal risk-taking is safe. Thereby, HCWs can develop their skills as well as their perception of psychological safety, improve their own health and well-being, and support the health of the patients they care for. Interpersonal communication in interdisciplinary settings could be an important input for the process of developing psychological safety, indicating that communication interventions can be beneficial to foster social relations and health.

High-quality care should also address individuals’ mental schemes of social relation in terms of psychological safety since this process is associated with the outcome of higher perceived patient safety. An open safety culture that includes speaking up against safety threats and learning from mistakes and adverse events could be achieved if psychological safety is improved on an individual and team level. Thus, error cultures and patient safety should be subject to team interventions focused on perspective-taking and mentalization on a regular basis.

Concretely, to improve effective communication to ensure the provision of high-quality patient care and improve patient safety in healthcare, interventions for individuals and teams could incorporate the following. These implications address psychological safety as a central facet of these processes.

Interpersonal communication skills training to communicate effectively. Such training can include learning active listening techniques, closed-loop communication, and speaking up or training clear, concise, and sufficient information exchange: Improved interpersonal communication competencies can help individuals and teams develop the skills needed to communicate openly and honestly with each other to create a shared understanding. This can lead to increased trust and respect within the team, which in turn can help create a safe and supportive environment where team members feel comfortable sharing their thoughts, feelings, and concerns [29,30,78]Standardized communication protocols can help reduce the risk of errors and misunderstandings in healthcare: Examples include standardized communication tools such as the SBAR (situation, background, assessment, recommendation) technique or implementing checklists to ensure important information is communicated effectively and consistently [31]. This is especially important in the face of the findings of the current research demonstrating that psychological safety is a key mediator.Feedback and debriefing sessions: Regular feedback and debriefing sessions can help individuals and teams reflect on their communication and identify areas for improvement [28,32]. This can be done through individual feedback sessions, team debriefs, or anonymous surveys to gather feedback from patients and colleagues.Creating a culture of learning and improvement: Team exercises can help create a culture of learning and improvement, where mistakes are viewed as opportunities for growth rather than failures. When team members feel safe making mistakes and learning from them, they are more likely to take risks and contribute their ideas [10,33,43].

In conclusion, healthcare should provide opportunities to develop interpersonal communication skills and reflective functioning training to improve safety performance indicators. Moreover, these results indicate that organizations should consider team psychological safety when assessing safety performance. Psychological safety is important to gain a better understanding of social relations and social resource promotion. If safety is not developed and shared cognitively, whether out of fear of punishment or relationship threat, patient safety concerns remain present [79].

## 7. Conclusions

In sum, the presented data highlight that interpersonal communication positively interrelates with perceived safety and seems to have an inhibiting effect on perceived threats. Moreover, these effects appear to be mediated by psychological safety. Our research demonstrates the importance of applying psychosocial processes and a systems psychodynamic perspective in a high-reliability organization (HRO) context such as (obstetric) healthcare to provide support where social factors (e.g., perceived interpersonal communication) are associated with social processes. This approach brings together advances in the domain of social and context relations by addressing psychological safety in correlation to patient safety. These findings can serve as an impetus for further research on psychosocial processes, psychodynamic theories and social and group determinants in a healthcare organization and improvements in practice. Accordingly, this research can contribute to the health and well-being of employees, enhancing social resources and interpersonal communication to complement strategies of healthcare and public health by means of focusing on the mediating factor of psychological safety. Concretely, high-quality care and patient safety can be ensured on the basis of this work by means of communication training as outlined above, addressing psychological safety.

## Figures and Tables

**Figure 1 ijerph-20-05698-f001:**
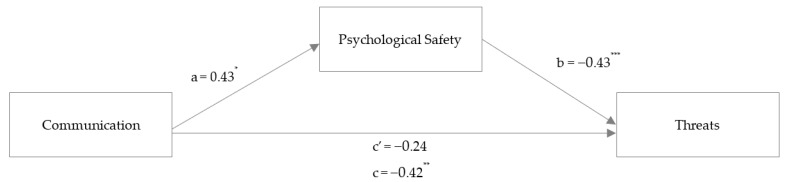
Study 1: Mediation analysis with the outcome variable threats. *Note*: mediation analysis in an online HCW sample (Study 1). Coefficients are reported as unstandardized regression coefficients for the relationship between communication and threats mediated by psychological safety. * *p* < 0.05, ** *p* < 0.01, *** *p* < 0.001.

**Figure 2 ijerph-20-05698-f002:**
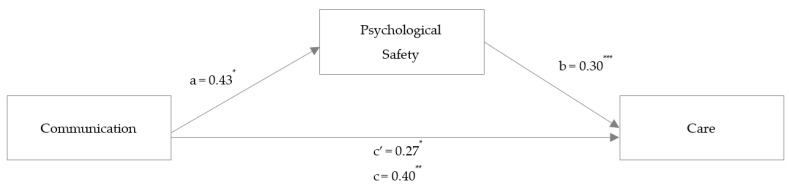
Study 1: Mediation analysis with the outcome variable care. *Note*: mediation analysis in an online HCW sample (Study 1). Coefficients are reported as unstandardized regression coefficients for the relationship between communication and care mediated by psychological safety. * *p* < 0.05. ** *p* < 0.01. *** *p* < 0.001.

**Figure 3 ijerph-20-05698-f003:**
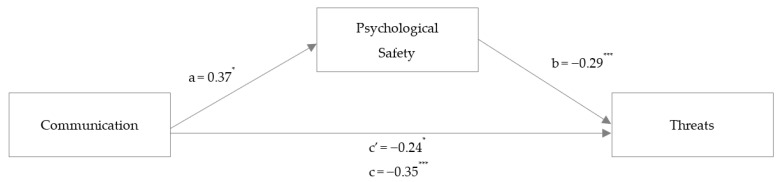
Study 2: Mediation analysis with the outcome variable threats. *Note*: mediation analysis in a hospital HCW sample (Study 2). Coefficients are reported as unstandardized regression coefficients for the relationship between communication and threats mediated by psychological safety. * *p* < 0.05, *** *p* < 0.001.

**Figure 4 ijerph-20-05698-f004:**
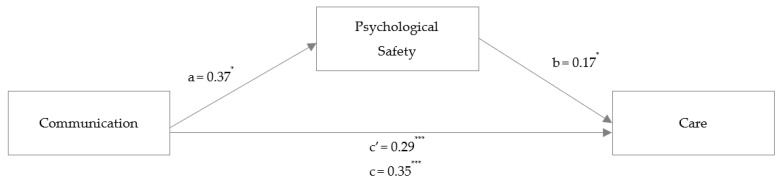
Study 2: Mediation analysis with the outcome variable care. *Note*: mediation analysis in a hospital HCW sample (Study 2). Coefficients are reported as unstandardized regression coefficients for the relationship between communication and care mediated by psychological safety. * *p* < 0.05, *** *p* < 0.001.

**Table 1 ijerph-20-05698-t001:** Study 1: Overview of socio-demographic data and experience among healthcare providers from the online survey.

	*N* = 129	Physicians (*n* = 18, 14%)	Midwives (*n* = 14, 11%)	Nurses (*n* = 58, 46%)	Other (Specified, e.g., Psychologist or Management, and Unspecified) (*n* = 24, 19%)
Sex	Women (*n* = 91, 71%)	9 (10%)	12 (14%)	42 (47%)	19 (21%)
	Men (*n* = 34, 26%)	8 (24%)	0 (0%)	16 (47%)	5 (15%)
	Divers (*n* = 1, 1%)	1 (100%)	0 (0%)	0 (0%)	0 (0%)
Age	<26 years (*n* = 8, 6%)	0 (0%)	1 (13%)	4 (50%)	2 (25%)
	26–40 years (*n* = 46, 36%)	7 (15%)	4 (9%)	26 (57%)	6 (13%)
	41–55 years (*n* = 46, 36%)	8 (18%)	5 (11%)	17 (38%)	10 (22%)
	>55 years (*n* = 26, 20%)	3 (12%)	2 (8%)	10 (40%)	6 (24%)
Experience	<1 year (*n* = 2, 2%)	0 (0%)	1 (50%)	0 (0%)	1 (50%)
	1–5 years (*n* = 23, 18%)	4 (17%)	4 (17%)	13 (57%)	2 (9%)
	>5 years (*n* = 103, 80%)	14 (14%)	9 (9%)	45 (45%)	21 (21%)

*Note*: frequencies and percentages are shown for each occupational group; percentages in parentheses. Up to 13 participants did not provide information on sex, age, and/or level of experience.

**Table 2 ijerph-20-05698-t002:** Study 1: Descriptive statistics and correlations from the online survey.

Variable		*M*	*SD*	1	2	3	4
1.	Communication Behavior	4.94	0.53				
2.	Psychological Safety	4.12	1.06	0.21 *			
3.	Patient Safety Threats	3.35	0.90	−0.22 *	−0.45 **		
4.	Quality of Care	4.09	0.75	0.31 **	0.51 **	−0.61 **	

Note: *N* = 128–129. * *p* < 0.05. ** *p* < 0.01.

**Table 3 ijerph-20-05698-t003:** Study 2: Overview of socio-demographic data and experience among healthcare providers from the hospital survey.

	*N* = 138	Physicians (*n* = 45, 33%)	Midwives (*n* = 45, 33%)	Nurses (*n* = 24, 18%)	Trainees (to Become Nurses or a Midwives) (*n* = 11, 8%)	Other (Specified, e.g., Psychologist or Management, and Unspecified) (*n* = 11, 8%)
Sex	Women (*n* = 125, 92%)	39(32%)	44 (36%)	22 (18%)	10 (8%)	9 (7%)
	Men (*n* = 11, 8%)	5 (46%)	1 (9%)	2 (18%)	1 (9%)	2 (18%)
Age	<26 years (*n* = 29, 22%)	1 (3%)	13 (45%)	3 (10%)	10 (34%)	2 (7%)
	26–40 years (*n* = 76, 58%)	36 (47%)	21 (28%)	15 (20%)	0 (0%)	4 (5%)
	41–55 years (*n* = 21, 16%)	4 (20%)	9 (45%)	3 (15%)	0 (0%)	4 (20%)
	>55 years (*n* = 6, 5%)	1 (17%)	1 (17%)	3 (50%)	0 (0%)	1 (17%)
Experience	<1 year (*n* = 21, 16%)	4 (19%)	7 (33%)	5 (24%)	4 (19%)	1 (5%)
	1–5 years (*n* = 57, 43%)	20 (35%)	21 (37%)	6 (11%)	6 (11%)	4 (7%)
	>5 years (*n* = 54, 41%)	19 (36%)	16 (30%)	12 (23%)	0 (0%)	6 (11%)

*Note:* Frequencies and percentages are shown for each occupational group; percentages in parentheses. Up to six participants did not provide information on sex, age, and/or level of experience.

**Table 4 ijerph-20-05698-t004:** Study 2: Descriptive statistics and correlations from the hospital survey.

Variable		*M*	*SD*	1	2	3	4
1.	Communication	4.55	0.65				
2.	Psychological Safety	4.32	0.89	0.31 **			
3.	Patient Safety Threats	3.00	0.73	−0.36 **	−0.43 **		
4.	Quality of Care	4.31	0.62	0.39 **	0.35 **	−0.45 **	

Note: *N* = 133–137. ** *p* < 0.01.

## Data Availability

The original data are not available for data protection reasons. However, requests can be addressed to the corresponding author.

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
