# Peer review of "Psychosocial Processes in Healthcare Workers: How Individuals’ Perceptions of Interpersonal Communication Is Related to Patient Safety Threats and Higher-Quality Care"

_ijerph, 2023, doi:10.3390/ijerph20095698_

Round 1

Reviewer 1 Report

The study on psychosocial processes in health workers: how individuals' perception of interpersonal communication is related to risks to patient safety and higher quality care is current and of interest to the scientific readers of this journal.

In order to increase the quality of the manuscript, the authors should provide the following information:

1. Make clear the difference between risk and threat for this study. This is because there is no analyzed risk matrix. Also, because the objective was to reduce threats (or risk as shown in the title).

2. Show some practical examples of interpersonal communication as a central form of social resources in the hospitals analyzed.

3. In the discussion section, clarify how high-quality care and patient safety are ensured with this work.

4. Show which are the communication and security performance indicators for this study. How do they vary with this study?

5. The approval was given by the Ethics Committee of the participating university (dated September 17, 2019). The data was collected from October 9, 2019 to March 6, 2020. Explain why until 2023 (three years later) it is submitted for publication?

6. It is necessary to compare the measurement scales:

a) In Reliability: they use Cronbach's alpha for internal consistency. They should compare with: test-retest or repeated tests

b) Comparative: use Likert. They should compare with: Thurstone.

c) With multi-item scales: such as Osgood and Stapel

Finally, the references must be updated. several are very old:

Ref 5, 24, 43: of 2006

Ref 11: from 2007

Ref 47 and 13 of 1999

Ref 17 and 27 of 1992

Ref 20 and 40 of 2000

Among many others

The writing quality is good

Author Response

Reviewer Feedback: 1. Make clear the difference between risk and threat for this study. This is because there is no analyzed risk matrix. Also, because the objective was to reduce threats (or risk as shown in the title).

***Author response: Thank you for feedback. We differentiated between threat and risk and revised the manuscript by cleary refering to threats (lines 47-57). 

Reviewer Feedback: 2. Show some practical examples of interpersonal communication as a central form of social resources in the hospitals analyzed.

***Author response: Thank you for feedback.We revised the manuscript and showed important practical examples for interpersonal communication as a central form of social resources in hospital settings and especially in obstetric units (as analyzed; lines 86-151).

Reviewer Feedback: 3. In the discussion section, clarify how high-quality care and patient safety are ensured with this work.

***Author response: Thank you for feedback. We revised the manuscript by clarifying how high-quality care and patient safety are ensured with this work (lines 603- 629). Concretely, we added the sentence “Concretely, high-quality care and patient safety can be ensured on the basis of this work by means of communication training as outlined above addressing psychological safety”.

Reviewer Feedback: 4.Show which are the communication and security performance indicators for this study. How do they vary with this study?

***Author response: Thank you for feedback. Communicaiton and safety performance indicators (operationalized as patient safety threats and quality care) are self-reported measurements of healthcare workers perceptions. To show, which are the communication and safety performance indicators, we added the items for the self reported perceptions in the supplementary material.

Reviewer Feedback: 5.the approval was given by the Ethics Committee of the participating university (dated September 17, 2019). The data was collected from October 9, 2019 to March 6, 2020. Explain why until 2023 (three years later) it is submitted for publication?

***Author response: Thank you for feedback.The present study was embedded in a larger research project, which was conducted over a duration of 4 years, however, the assessments were stopped due to the covid19-pandemic. For reasons of practicability and prioritization, data analysis and work on publications began in the last project phases after also the other phases were finished,namely during the dissemination phase.

Reviewer Feedback: 6. It is necessary to compare the measurement scales:

  1. a) In Reliability: they use Cronbach's alpha for internal consistency. They should compare with: test-retest or repeated tests
  2. b) Comparative: use Likert. They should compare with: Thurstone.
  3. c) With multi-item scales: such as Osgood and Stapel

***Author response: Thank you for this suggestion. We used Cronbach's alpha because this is a statistical measure which is used to evaluate the internal consistency of a scale (Cronbach, 1951; Taber, 2018). It is a commonly used method for assessing the reliability of tests or surveys which is a central indicator of the measurement quality (DeVellis, 2017). We are more interested in practical aspects and with that Cronbach's alpha can help researchers or practitioners to determine if a scale or set of related items is measuring what it is intended to measure in a consistent and reliable manner (Taber, 2018). A higher value of Cronbach's alpha indicates that the scale is more reliable and consistent, which means that the results are more accurate and trustworthy (Cronbach, 1951). With the values we calculated for our measures, we can conclude that the measures perform satisfactory.

We agree that it is important to note that Cronbach’s alpha is just one measure of reliability and should ideally be used in conjunction with other methods to ensure the validity of the measurement tool. Additionally, Cronbach’s alpha can be affected by factors such as the number of items in the scale, the nature of the items, and the sample size, among other aspects (De Vaus, 2014). However, test-retest or repeated tests are a method used to assess the internal consistency of a measure by comparing the scores of the same individuals on the same test taken at two different points in time. While this method has some advantages, it also has several limitations and inadequacies (Carifio, & Perla, 2008). For instance:

*Practice effects: Participants may perform better on the second administration of the test due to practice or familiarity with the test items. This could lead to an overestimation of the internal consistency of the measure.

*Memory effects: Participants may remember their responses from the first administration of the test, which could bias their responses on the second administration of the test. This could also lead to an overestimation of the internal consistency of the measure.

*Time interval: The time interval between the two administrations of the test could affect the results. If the time interval is too long, there may be changes in the participants’ knowledge, skills, or attitudes that could affect their performance on the test.

*Regression to the mean: Participants who perform very well or very poorly on the first administration of the test are likely to perform closer to the mean on the second administration, which could affect the estimates of the internal consistency of the measure.

*Attrition: Participants may drop out or be lost to follow-up between the two administrations of the test, which could introduce bias into the results.

*Restriction of range: Participants who perform very well or very poorly on the first administration of the test may be less likely to show variability on the second administration, which could affect the estimates of the internal consistency of the measure.

Overall, while test-retest or repeated tests can be a useful method for assessing the internal consistency of a measure, we see more limitations and potential inadequacies and to use it in conjunction with other methods to ensure the validity of the measure. Thus, we rather stick to the former procedure with just keeping Chronbach’s Alpha. The same holds true with Thurstone, Osgood and Stapel: Thurstone, Osgood, and Stapel are three methods used to assess internal consistency or reliability of a measure (Crano & Prislin, 2011). While these methods have some advantages, they also have several limitations and inadequacies (Taber, 2018). Here are some of them:

*Subjectivity: All three methods involve subjective judgments by raters or judges, which can introduce bias into the results. The subjective nature of these methods means that the results can vary depending on the rater or judge who is doing the rating.

*Limited scope: Thurstone and Osgood focus on a limited number of attributes or dimensions of the construct being measured, while Stapel only measures one attribute. This can make it difficult to capture the full range of variation in the construct being measured.

*Difficulty in interpretation: The results of Thurstone and Osgood can be difficult to interpret because they involve complex statistical procedures, such as factor analysis, that are not always easy to understand. The results of Stapel are easier to interpret but can be limited by the fact that they only measure one attribute.

*Limited generalizability: Thurstone, Osgood, and Stapel are designed to assess internal consistency or reliability of measures within specific populations or contexts. This can limit their generalizability to other populations or contexts.

*Time-consuming: All three methods can be time-consuming and require a significant amount of resources, which can make them impractical for use in large-scale studies or surveys.

*Lack of validity: While internal consistency or reliability is an important aspect of measuring a construct, it is not sufficient for establishing validity. Thurstone, Osgood, and Stapel do not address issues of validity, which can limit their usefulness in assessing the overall quality of a measure. Here the references, we consulted and aggree with:

Crano, W. D., & Prislin, R. (2011). Attitudes and attitude change. Psychology Press.

Cronbach, L. J. (1951). Coefficient alpha and the internal structure of tests. Psychometrika, 16(3), 297-334.

DeVellis, R. F. (2017). Scale development: Theory and applications (4th ed.). SAGE Publications.

Haws, K. L., Sample, K. L., & Hulland, J. (2023). Scale use and abuse: Towards best practices in the deployment of scales. Journal of Consumer Psychology, 33(1), 226-243.

Tavakol, M., & Dennick, R. (2011). Making sense of Cronbach's alpha. International Journal of Medical Education, 2, 53-55.

De Vaus, D. A. (2014). Surveys in social research (6th ed.). Routledge.

Nunnally, J. C., & Bernstein, I. H. (1994). Psychometric theory (3rd ed.). McGraw-Hill.

Carifio, J., & Perla, R. (2008). Resolving the 50-year debate around using and misusing Likert scales. Medical Education, 42(12), 1150-1152.

Thurstone, L. L. (1928). Attitudes can be measured. American Journal of Sociology, 33(4), 529-554.

Osgood, C. E. (1952). The nature and measurement of meaning. Psychological Bulletin, 49(3), 197-237.

Stapel, D. A., & Koomen, W. (2001). The use of factor analysis in the development of Stapel scales. European Journal of Psychological Assessment, 17(3), 212-221.

Taber, K.S. (2018). The Use of Cronbach’s Alpha When Developing and Reporting Research Instruments in Science Education. Res Sci Educ 48, 1273–1296. https://doi.org/10.1007/s11165-016-9602-2

Overall, we did not add any additional statistics, as we assume our approach to be sufficient and best matched to the needs of the readers.

Reviewer Feedback: Finally, the references must be updated. several are very old:

Ref 5, 24, 43: of 2006

Ref 11: from 2007

Ref 47 and 13 of 1999

Ref 17 and 27 of 1992

Ref 20 and 40 of 2000

Among many others

***Author response: Thank you for your comment. We updated several old refrences, nonetheless some milestone publications (e.g., Edmondson´s psychological safety literature) still used, regardless of the published year.

Reviewer 2 Report

The manuscript consists of total 19 pages, including 4 tables, 4 figures and the list of total 64 literature references. The article presents the results of 2 studies aiming at estimating the role of social processes, communication and interaction with and among healthcare workers in the attempts of improvement in dealing with the risks to the safety of the patients. As such, the article discusses one of current and important problems. The title is informative and clear enough, relevant to the contents of the main text. The English quality of the text is good, however here are some text-edition flaws in the form of multiple "Klicken oder tippen Sie hier, um Text einzugeben" statements [e.c. line 76,88, 98], which shall be deleted. The text has a logical structure and the Authors' line of argumentation is easy to follow.

The Abstract is not structured but it mirrors the key contents of the main text.

The Introduction section provides extensive information on theoretical and practical background behind the justification of the study.

The Material and methods are described in high detail for both presented studies.

The Results are consistent with the declared methodology and clearly presented for both studies.

The Discussion fits the Authors' study results into the broader context of previously published knowledge, pointing at practical aspects as well.

The Tables and Figures are adequately chosen to support the Authors' line of presenting the data.

The Conclusions are based on the discussed results and careful enough. Although the abbreviation HRO was explained earlier in the text, it is advisable to explain it in the Conclusions section once again for the use of those Readers who tend to jump right to the conclusions before deciding whether or not to go through the main text of the article.

The Literature References are numerous and reasonably recent.

Author Response

Reviewer Feedback: The Abstract is not structured but it mirrors the key contents of the main text.

The Introduction section provides extensive information on theoretical and practical background behind the justification of the study.

The Material and methods are described in high detail for both presented studies.

The Results are consistent with the declared methodology and clearly presented for both studies.The Discussion fits the Authors' study results into the broader context of previously published knowledge, pointing at practical aspects as well.

The Tables and Figures are adequately chosen to support the Authors' line of presenting the data.

***Author response: Thank you for your positive feedback and for taking the time to review the manuscript. We appreciate the time and effort. For the abstract, we checked whether we adhered to the author guidelines: “The abstract should be a single paragraph and should follow the style of structured abstracts, but without headings”. Thus, we did not add headings but worked on improving it so that it is clearly structured.

Reviewer Feedback: The Conclusions are based on the discussed results and careful enough. Although the abbreviation HRO was explained earlier in the text, it is advisable to explain it in the Conclusions section once again for the use of those Readers who tend to jump right to the conclusions before deciding whether or not to go through the main text of the article.

***Author response: Thanks for the advice. We have revised the Discussion and Conclusion section and explained the abbreviation of HRO for reasons of comprehensibility and readability. The text now either fully states the term high reliability organizations or explains the abbreviation (line 528; line 563f; line 735).

Reviewer 3 Report

I congratulate the authors for the topic chosen for this research. The manuscript addresses a relevant subjet; the relationship between interpersonal communication in the healthcare team, personal satisfaction and risks to the patient.

The introduction correctly situates the topic of study.

The methodology used is adequate and clearly explained.

In Study 2, the sample used reveals some doubts about the generalizability of the results obtained. Although this circumstance is reflected in the limitations of the study, it would help if the total number of professionals that make up the obstetrics services of the participating hospitals were reflected, differentiating the professional categories.

The results are very complete and ratify the hypotheses put forward.

The discussion is well developed and the conclusions are clear.

I encourage the authors to continue research in this line of investigation with a qualitative study on interpersonal relationships and communication in multidisciplinary health care groups.

Author Response

Reviewer Feedback: The introduction correctly situates the topic of study.

The methodology used is adequate and clearly explained.

***Author response: Thank you for your positive feedback! We appreciate the time and effort for reviewing this manuscript.

Reviewer Feedback: In Study 2, the sample used reveals some doubts about the generalizability of the results obtained. Although this circumstance is reflected in the limitations of the study, it would help if the total number of professionals that make up the obstetrics services of the participating hospitals were reflected, differentiating the professional categories.

***Author response: Thank you for feedback. Study 2 was performed within a research project concerning digitally supported communication in obstetrics and gynecology. The research project aimed to conduct a full survey, therefore the majority of the sample (N =138 out of N =141) who worked at the hospitals were participating in the reseach project (lines 421-423).  Regarding the generalizability, the study design realized to have two representative hospitals from Germany participating in the study. Thus, the findings should generally be generazible to other hospitals but we agree that within the hospitals, there might be a selective sample. We have adapted the discussion accordingly.

Reviewer Feedback: The results are very complete and ratify the hypotheses put forward.The discussion is well developed and the conclusions are clear.

I encourage the authors to continue research in this line of investigation with a qualitative study on interpersonal relationships and communication in multidisciplinary health care groups.

***Author response: Thank you for feedback and the encouraging research idea. We integrated your suggestion as as future research possibility. The text now reads “Moreover, an investigation with a qualitative study in multidisciplinary healthcare groups could provide new insights into the relationship between interpersonal relationships and communication, as we have done it successfully before [e.g., 27] but which should be done more often in the future, too“ (lines 653-656).

Reviewer 4 Report

The article "Psychosocial Processes in Healthcare Workers: How Individuals’ Perception of Interpersonal Communication is Related to Patient Safety Risks and Higher Quality Care" has in its scope a well-defined object of study, with research questions and well-defined hypotheses. The article's theme is also interesting, as it addresses the psychosocial process of health workers and patient safety from an innovative and relevant perspective.

The introduction contextualizes the problem and the object of study of the research well, and the objective of the work is clear.

The methodology is well founded, the results presented are good. The answer to the research questions and the tests of the hypotheses were described in a statistically adequate way. Therefore, it was possible to satisfactorily qualify the scientific findings of the research.

The presented limitations demonstrate that there are still possibilities for other researches in this same theme, which could improve the presented scientific findings and future works.

This is an article of good scientific quality. However, I make two recommendations to authors:

[1] Include a section of related works, there is no need to be a very extensive section, but it should point out other works that are related to the theme. This section should make a critical analysis of the works, the objective is to situate the research and demonstrate more clearly the contributions of this research in relation to the others mentioned.

[2] Authors should delve further into the discussion section and point out the importance of this research to patient safety. How to have a more favorable environment for communication and for the exercise of quality in intersubjective relationships? This question could be answered in the discussion.

I also recommend a revision of the introduction, just to avoid redundancy in some parts of the text.

The article has good writing quality, but I recommend that a general revision of the text be made for the final version.

Author Response

Reviewer Feedback: The introduction contextualizes the problem and the object of study of the research well, and the objective of the work is clear.

The methodology is well founded, the results presented are good. The answer to the research questions and the tests of the hypotheses were described in a statistically adequate way. Therefore, it was possible to satisfactorily qualify the scientific findings of the research.

The presented limitations demonstrate that there are still possibilities for other researches in this same theme, which could improve the presented scientific findings and future works.

***Author response: Thank you for your positive feedback and for taking the time to review the manuscript!

Reviewer Feedback: This is an article of good scientific quality. However, I make two recommendations to authors:

[1] Include a section of related works, there is no need to be a very extensive section, but it should point out other works that are related to the theme. This section should make a critical analysis of the works, the objective is to situate the research and demonstrate more clearly the contributions of this research in relation to the others mentioned.

***Author response: Thank you for feedback. We incluced a section of related works, which points out the contributions of our research in relation to other important healthcare studies in the field of communication, social relations and patient safety. We outline that the relationship between communication errors and patient safety has been much studied, yet the mechanisms of this relationship have not been clearly explored. Thus, our reseach model which depicts research into this mechanism is important to derive further patient safety measures and to improve social resources such as communication and collaboration (lines 181-194). Also, we added a long section on the practical examples of interpersonal communication as a central form of social resources in hospitals in general. With these examples, related works is mentioned more explicitly (lines 86-151).

Reviewer Feedback: [2] Authors should delve further into the discussion section and point out the importance of this research to patient safety. How to have a more favorable environment for communication and for the exercise of quality in intersubjective relationships? This question could be answered in the discussion.

***Author response: Thank you for feedback. We revised the manuscript and pointed out the importance of our reseach to patient safety and how to foster more favorable environment for communication and for the exercise of quality in intersubjective relationships (lines 600-629).

Reviewer Feedback: I also recommend a revision of the introduction, just to avoid redundancy in some parts of the text.

***Author response: Thank you for feedback. We revised the introduction in oder to avoid redundancies (lines 44-194)

Reviewer Feedback: The article has good writing quality, but I recommend that a general revision of the text be made for the final version.

***Author response: Thank you for your comment. The revised manuscript was proofread by all co-authors and the final version was also be checked by a native speaker which we indicate with the acknowledgments: “We appreciate the help of Mathew Perez with proofreading this manuscript”.

Round 2

Reviewer 1 Report

The authors have applied and corrected each reviewer's comment, which increases the quality of the manuscript.

Minor editing of English language required